# Trans-Mucosal Efficacy of Non-Thermal Plasma Treatment on Cervical Cancer Tissue and Human Cervix Uteri by a Next Generation Electrosurgical Argon Plasma Device

**DOI:** 10.3390/cancers12020267

**Published:** 2020-01-22

**Authors:** Thomas Wenzel, Daniel A. Carvajal Berrio, Christl Reisenauer, Shannon Layland, André Koch, Diethelm Wallwiener, Sara Y. Brucker, Katja Schenke-Layland, Eva-Maria Brauchle, Martin Weiss

**Affiliations:** 1Department of Women’s Health, Eberhard Karls University, 72076 Tübingen, Germany; thomas.wenzel@student.uni-tuebingen.de (T.W.); Daniel.Carvajal-Berrio@med.uni-tuebingen.de (D.A.C.B.); christl.reisenauer@med.uni-tuebingen.de (C.R.); shannonlayland@yahoo.com (S.L.); Andre.Koch@med.uni-tuebingen.de (A.K.); Diethelm.Wallwiener@med.uni-tuebingen.de (D.W.); sara.brucker@med.uni-tuebingen.de (S.Y.B.); katja.schenke-layland@uni-tuebingen.de (K.S.-L.); eva-maria.brauchle@uni-tuebingen.de (E.-M.B.); 2Cluster of Excellence iFIT (EXC 2180) “Image-Guided and Functionally Instructed Tumor Therapies”, Eberhard Karls University, 72076 Tübingen, Germany; 3Natural and Medical Sciences Institute (NMI), 72770 Reutlingen, Germany; 4Department of Medicine/Cardiology, University of California, Los Angeles (UCLA), Los Angeles, CA 90095, USA

**Keywords:** tissue penetration, non-thermal plasma, non-invasive plasma treatment (NIPP), cervical intraepithelial neoplasia (CIN), Raman imaging, Raman microspectroscopy, Plasma lipid interactions

## Abstract

Non-invasive physical plasma (NIPP) generated by non-thermally operated electrosurgical argon plasma sources is a promising treatment for local chronic inflammatory, precancerous and cancerous diseases. NIPP-enabling plasma sources are highly available and medically approved. The purpose of this study is the investigation of the effects of non-thermal NIPP on cancer cell proliferation, viability and apoptosis and the identification of the underlying biochemical and molecular modes of action. For this, cervical cancer (CC) single cells and healthy human cervical tissue were analyzed by cell counting, caspase activity assays, microscopic and flow-cytometric viability measurements and molecular tissue characterization using Raman imaging. NIPP treatment caused an immediate and persisting decrease in CC cell growth and cell viability associated with significant plasma-dependent effects on lipid structures. These effects could also be identified in primary cells from healthy cervical tissue and could be traced into the basal cell layer of superficially NIPP-treated cervical mucosa. This study shows that NIPP treatment with non-thermally operated electrosurgical argon plasma devices is a promising method for the treatment of human mucosa, inducing specific molecular changes in cells.

## 1. Introduction

Cold atmospheric plasma (CAP) has offered promising anti-neoplastic effects on pancreatic, prostatic and gynecological tumors, as well as melanoma and glioma [1,2,3,4,5]. Cold or “non-thermal” plasmas are not in thermodynamic equilibrium due to the temperature of electrons being much hotter than the temperature of the atomic nuclei. Thermalized electrons, however, show a non-Maxwellian velocity distribution, resulting in plasma temperatures being adjustable to body temperature [6,7,8]. Most importantly, the interaction of the reactive plasma factors with gaseous, liquid or solid substances is followed by the generation of reactive oxygen and nitrogen species (ROS and RNS), which is responsible for the induction of anti-proliferative cell mechanisms and cancer cell death [6,7,8,9,10]. To date, there are only few medically approved CAP devices. Thus, the investment and operation costs for these devices are relatively high. Similar to conventional CAP devices, the plasma effluent of non-thermally operated electrosurgical argon plasma sources contains diverse biologically reactive factors (charged particles and molecules, free radicals, ultraviolet, and infrared radiation) [11]. Commonly used electrosurgical argon plasma sources are devices for high-frequency (HF)-based surgery. These plasma sources are clinically well established and have been available for various clinical procedures for many years [12]. A great advantage of these devices is the variety of possible clinical applications using the highly flexible and sterile application probes in open and endoscopic surgery, which are associated with relatively low overheads. The system contains two electrodes. One electrode is active, and the other electrode is a neutral “ground”, usually placed on the outer skin surface with proximity to the treated body region. The thermal effect during tissue coagulation is based on the contactless transfer of high-frequency alternating current from the plasma probe to the target tissue via ionized electrically conductive argon plasma, followed by energy conversion within the tissue (Joule heating) [13]. The argon plasma corridor thereby follows the way of lowest electrical resistance. The thermal effect is directly proportional to the electrical resistance of tissue and the square of amperage and can be avoided by using low energies and performing continuous motion of the plasma probe [11]. However, usually inducing thermal tissue effects, the ignited argon plasma of next generation electrosurgical argon plasma sources, as utilized in this study, is discussed to comprise lower energy per plasma particle as well as to combine characteristics of cold, non-equilibrium plasma, since thermal equilibrium within the plasma would require multiple actually used currents [14,15,16,17]. Due to the divergent principle of non-thermal plasma generation by electrosurgical argon plasma devices, we use the term non-invasive physical plasma (NIPP) treatment.

Local NIPP treatment is a promising therapeutic procedure for various chronic inflammatory and neoplastic diseases of human mucosa, such as cervical intraepithelial neoplasia (CIN), where current treatment strategies are highly invasive, painful and associated with serious short- and long-term side effects and risks, especially during pregnancy [18,19,20,21,22,23,24].

Very few in vitro studies have investigated the effects on cancer cells and tissue penetration during NIPP treatment on human mucosa [11,25]. Previously, our group showed the feasibility of non-thermal treatment on human tissue using electrosurgical argon plasma to significantly increase the potential of generating free radicals. Cancerous and precancerous diseases often originate from the basal cell layer of the epithelium; therefore, it is crucial to evaluate the tissue penetration and the effects of NIPP treatment within human tissues.

In this study, we aimed to investigate cervical cancer (CC) cell proliferation and cell death after NIPP treatment, compared to primary cells from healthy cervical tissue. Moreover, we employed Raman microspectroscopy to identify distinct lipid-based molecular changes in cells after NIPP treatment and to track the penetration of NIPP effects in human mucosal tissues derived from the cervix uteri. Raman microspectroscopy is a laser-based technique, which excites molecular vibrations in a sample to reflect the tissue’s specific molecular and biochemical composition [26].

NIPP treatment caused significant antiproliferative and apoptotic effects, particularly in CC cells associated with cell membrane damage. Raman microspectroscopy and imaging showed significant alterations of lipids’ molecular composition and morphological differences in lipids after NIPP exposure.

## 2. Results

### 2.1. Antiproliferative Cell Response after NIPP Treatment

The impact of NIPP exposure on cell proliferation and cell death in human CC cell line SiHa and primary cells from healthy cervical tissue of three independent donors (each independent experiment was performed with cells from a different donor) was assessed based on cell numbers, brightfield microscopy, flow cytometry, and Caspase-Glo 3/7 luminescent assay. NIPP treatment showed a dose-dependent decrease in cell proliferation within 120 h compared to argon-gas treated controls with the same initial cell number (Figure 1a,b). In line with this, confocal microscopy revealed cell depletion, accompanied by altered cell morphology and signs of coarsening (Figure 1c). Flow cytometry showed a significant decrease in cellular viability, from 20% to 40%, in both cell entities (Figure 1d,e). Beside the decrease in cellular viability, a significant activation of caspases could be determined, utilizing the Caspase-Glo 3/7 luminescent assays, which measured the activation of the prominent effector caspases 3 and 7 (Figure 1f,g). In CC cells, this effect was highly significant at all investigated time points; however, non-cancerous cells showed less strong caspase activation.

SiHa cells and non-cancerous cells were treated with NIPP or argon-gas in six-well cell culture dishes and analyzed by live/dead staining with propidium iodide (PI) and fluorescein diacetate (FDA) and fluorescence microscopy after the indicated time points. Figure 2 shows representative fluorescence microscopic images and the respective software-based biological image analysis of three independent experiments. The cells were NIPP treated for 30 s and analyzed within 120 h. NIPP exposure resulted in decreased cell numbers, an increase in dead cells, represented by red PI stained cell nuclei (Figure 2a,c), and significantly decreased cell viability, calculated by the ratio of PI positive to PI negative/FDA positive cells. Compared to flow cytometry (Figure 1b,d), which analyzed the cell suspension after trypsinization, live/dead staining solely measured attached cells. Thus, the loss of mostly detached cells previous to live/dead staining resulted in a relative reduction in NIPP effects. Figure 2e schematically illustrates the mechanism of FDA and PI staining in viable and dead cells. As a cell membrane permanent adduct, FDA needs activation through an intact enzymatic esterase activity and full integrity of the cell membrane to enable its intracellular arrest, whereas PI can only pass through porous cell membranes before the nuclear staining of dead cells. In both cancerous and non-cancerous cells, PI/FDA agents therefore strongly point towards NIPP-induced cell membrane damage and molecular changes in lipids, either as (i) a direct result of cellular NIPP exposure or (ii) a secondary effect within the propagation of programmed cell death.

### 2.2. NIPP Induced Changes of Lipids in Human Cervical Mucosa Identified by Raman Imaging

Utilizing Raman microspectroscopy and multivariate data analysis, we biochemically characterized lipids as molecular components in human cervical tissues (Figure 3). The fresh, primary tissue samples were superficially NIPP-treated for 2 and 5 min without the generation of tissue harming heat effects, characterized by surface temperatures of around 22.8 °C during 2 min of tissue treatment (Figure 4).

Cryosections of NIPP-treated tissue and argon controls were subsequently analyzed by Raman microspectroscopy immediately or after 24 h of incubation. We used true component analysis (TCA) and established spectral bands (e.g., 2850 rel. cm^−1^, as previously described [26]) to identify lipid tissue components within the analyzed cervical epithelium (Figure 3). Lipids [27,28] (green) were identified as characteristic and very consistent components of the superficial and basal epithelial cell layers (Figure 5a). The cervical epithelium is a highly proliferative and important histological structure. Here, the multi-layered hornless squamous epithelium changes into single-layered mucus-forming cylindrical epithelium. While cell proliferation takes place in the basal cell layer, cells significantly differentiate and change morphology on their way into the superficial tissue layers to fulfill important tasks for tissue integrity and antiseptics. Comparable to Movat Pentachrom tissue staining, the Raman image showed a superficial epithelium, characterized by a loose composite of lipid-rich tissue and a basal epithelial area, characterized by a highly organized lipid-rich and palisade-like membrane architecture. Also, the parabasal and intermediate tissue structures of the Raman images were highly comparable to the histochemical staining. Due to the histological and biochemical differences between the superficial and basal tissue layers, the lipid composition of superficial layers was only compared to superficial, and basal layers only with basal in all the following experiments (for further details please see Wenzel et al., 2019) [26].

Highly specific bands for lipids (e.g., 2850 rel. cm^−1^) enabled the analysis of the lipid distribution and lipid peak intensities in plasma-treated tissues. According to Raman images, NIPP treatment resulted in slightly increased peak intensities of the molecular lipid components (Figure 5b). However, this effect could not be demonstrated as statistically significant (*p*-value: sup. 0 h: 0.66, 24 h: 0.38, bas. 0 h: 0.83, 24 h: 0.13). Raman imaging showed visible changes in the morphology of lipid components in NIPP-treated tissues, mainly represented by coarsening and rough clumped structural rearrangements.

Principle component analysis (PCA) of Raman spectra representing the lipid components revealed relevant biochemical differences between the NIPP-treated and argon control samples (Figure 5c). The loadings of the respective scores indicated plasma-dependent differences at characteristic spectral positions linked to lipid components (1169, 1306, 1368 and 2910–2920 rel. cm^−1^) (Figure 5d) [29,30,31,32]. All of the detected wavenumbers of characteristic Raman peaks for the lipid components are summarized in Table 1.

Relevant and statistically significant differences were found between the plasma- and argon control-treated samples (each *n* = 3) by analyzing the PCA score values of the lipid spectra for all indicated time points and parameters (Figure 5c). NIPP treatment resulted in immediate and consistent lipid effects in the superficial layers (∆ 0 h: 2 min = 0.10, 5 min = 0.14; ∆ 24 h: 2 min = 0.09, 5 min = 0.11). Meanwhile, immediate plasma–lipid effects in the basal tissue layers were much lower (∆ 0 h: 2 min = 0.01, 5 min = 0.06), but still statistically significant. The NIPP-dependent effects on lipid components reached levels comparable to those of the superficial tissue layers after 24 h of incubation (∆ 24 h, 2 min = 0.05, 5 min = 0.10). Compared to the increasing NIPP-effect on lipids in basal cell layers, the immediate and strong effect in superficial tissue layers decreased within 24 h. The effects of NIPP on lipid components were all statistically highly significant (each *p* < 0.0001) independent of the tissue layers and the indicated incubation times.

## 3. Discussion

Most devices in development for the in vivo treatment of human skin and mucosa with non-thermal physical plasma are cold plasma devices, based on dielectric barrier discharge (DBD) or atmospheric pressure plasma jet (APPJ) technology. However, plasma treatment by non-thermally operated electrosurgical argon plasma sources are gaining importance as they show various advantages. Common electrosurgical argon plasma sources are available in many clinics and have been established for clinical use for many years. Due to highly flexible and sterile application probes with various possible clinical applications, these devices are associated with relatively low costs. Nevertheless, little is known about the non-thermal effects of these devices in human cancer and solid epithelial tissue. Human mucosa is the main target tissue for future plasma-based treatment procedures of many chronic inflammatory, precancerous and cancerous diseases [33]. In previous work, we characterized the principal ability of electrosurgical argon plasma sources to perform non-thermal plasma treatment of human tissue, referred to as NIPP [11]. In this study, we demonstrated that dynamic tissue treatment by a next generation electrosurgical plasma source (VIO^®^ 3, APC 3) was not associated with potentially tissue harming surface temperatures (Figure 4). Moreover, the generation of reactive oxygen and nitrogen species (RONS), due to the interaction of the plasma effluent with both liquid and solid biological interfaces, was significantly higher in case of electrosurgical argon plasma devices [11]. Radical “trapping” and electron-spin-resonance spectroscopy of isotonic liquids and human mucosa showed the significant induction of •OH and •H radicals in liquids, whereas in human tissue, the carbon-centered radicals were most abundant. RONS-dependent apoptosis is by far the most evident cellular mechanism of plasma treatment in the literature, particularly for malignant cell entities [34]. In line with this, NIPP treatment of suspended single cells significantly decreased proliferation and viability and increased apoptotic cell mechanisms in cancerous and non-cancerous cells of the cervix uteri, measured by cell counting, FACS and Caspase 3/7 activity assay (Figure 1, Figure 2 and Figure 3). However, the induction of apoptosis was higher in CC cells compared to cells from healthy cervical tissue. The less significant apoptotic effect on non-cancerous cells is in line with an increasing number of studies utilizing APPJ, including our group [35,36]. Single cells were treated in suspension to avoid mechanical detachment and associated cell damage, as well as drying effects. The unphysiological suspension state was limited to the treatment period before enabling the immediate reattachment of the cells. We performed indirect plasma treatment via plasma-activated liquid (PAL) in this procedure, which was recently shown to reveal very similar anti-proliferative cell effects compared to direct plasma treatment [3].

Recent studies demonstrated the important role of plasma-dependent damage to cell membranes and RONS-mediated tissue lipid peroxidation for the induction of antiproliferation and cell death [37,38]. To correlate the membrane integrity after NIPP treatment with increased cell death, we performed live/dead staining with PI and FDA. For fluorescence activation, FDA is dependent on intact intracellular enzymatic activity as well as cell membrane integrity to sufficiently accumulate. PI exclusively passes through porous cell membranes to enter the nuclei. We found significant occurrence of PI-positive nuclei, which could be the result of apoptosis-related (late) cell membrane fragmentation or immediate plasma-induced membrane damage.

Raman microspectroscopy and multivariate data analysis have been used for the analysis of various types of human cells and tissues and for the biochemical characterization of different molecular components after APPJ tissue treatment [26,39,40,41,42]. Here, we evaluated the impact of NIPP treatment on lipids in solid cervical tissue. Raman imaging of solid tissue samples enabled the simultaneous generation of morphological and biochemical information after tissue treatment. This is a great advantage compared to previous studies, which aimed to analyze plasma’s effects on lipids in the outermost skin layer of keratinized epithelium by Raman microspectroscopy [43]. Further studies demonstrated the ability of Raman spectroscopy to identify plasma-driven changes in lipids and lipid droplets (composed of phospholipids and triglycerides) in bacterial spores, budding-yeast and HeLa cells [44]. Previously, we demonstrated that Raman imaging can identify significant molecular differences in cells belonging to the either the basal or superficial layer of native epithelial tissue [26]. Therefore, in this study, we only compared superficial layers with superficial, and basal layers with basal, respectively. The assessment of the epithelium showed that NIPP treatment has an immediate impact on the biochemical composition and morphology of lipids in the superficial as well as basal tissue layers (Figure 5). We could demonstrate that these effects were clearly dose- (difference between 2- and 5-min treatment) and incubation time-dependent (difference between 0 and 24 h incubation). The peaks at 2850 (ν_s_ CH_2_, lipids, fatty acids, CH_2_ symmetric), 2880 (CH_2_ asymmetric stretch of lipids and proteins) and 2950 rel. cm^−1^ (CH_3_ asymmetric stretch) showed that the lipid signal is higher in the argon-treated control [29]. Superficial NIPP treatment of the epithelial tissue for 2 and 5 min of was followed by relevant changes in the biochemical lipid composition. In particular, these changes were highly significant for 5 min (Figure 5). Previously, further human studies investigated 5 min plasma treatment, without inducing relevant side effects [45,46]. In this study, 5 min of superficial NIPP treatment was followed by highly significant effects on lipid molecules in the basal cell layer but did not induce mucosal damage under the ex vivo conditions. Previously, our group characterized the plasma tissue penetration depth in non-keratinous human mucosa of the APPJ kINPen med [26]. Thereby, the specific molecular effects on DNA were used to track the plasma penetration into the basal cell layer, complementary to a penetration depth of 270 µm. In the present study, NIPP-mediated changes in lipids could be specifically identified in the basal cell layer, suggesting that NIPP-generated reactive species efficiently transmigrate through the full thickness of human mucosa.

## 4. Materials and Methods

### 4.1. Cell Culture

Cervical squamous cell carcinoma-derived and human papillomavirus (HPV)-positive SiHa cells were purchased from ATCC (ATCC^®^ TCP-1022™, American Type Culture Collection, Manassas, VA, USA).

Primary cells from healthy cervical tissue were isolated from 3 different donors after surgical removal of the cervix uteri at the Department of Women’s Health, University Hospital Tübingen, Germany. The scientific use of human tissue samples was approved by the institutional review board of the medical faculty of the University Hospital Tübingen (ethical vote: 649/2017BO2). Written informed consent was obtained from all patients. The tissue samples were transported in Dulbecco’s modified eagle’s medium (DMEM), supplemented with 1% penicillin/streptomycin. To confirm the benign nature of the primary tissue, pathological review was performed by a gynecological pathologist at the pathology department of the University Hospital in Tübingen. For primary cell isolation, surgically-removed tissues were cut into pieces of 1–2 mm and washed with PBS. After incubation with trypsin/EDTA for 30 min, the surface of the tissue pieces was scraped off and filtered through a cell sieve. Hereinafter, healthy primary cells from the cervix uteri were denoted as non-cancerous cells. The results describe an average of three independent experiments, with each being performed using cells from an independent donor.

SiHa and non-cancerous cells were cultured in Dulbecco’s modified eagle’s medium (DMEM F12, Cat. no. 11320033, Fischer Scientific, Waltham, MA, USA), supplemented with 10% fetal calf serum (Life Technologies, Carlsbad, CA, USA), 1 mM sodium pyruvate (Life Technologies) and 1% penicillin/streptomycin (Invitrogen, Carlsbad, CA, USA) at 37 °C and 5% CO2 in a humidified atmosphere. Every 2–3 days, a media exchange was performed, and cells were passaged after reaching 70%–80% confluence. The adherent cells were detached by 0.25% trypsin-EDTA (Life Technologies).

### 4.2. Plasma Treatment

To generate NIPP, utilizing argon as a carrier gas, we used the electrosurgical device VIO^®^ 3, APC 3 (Erbe Elektromedizin, Tübingen, Germany) and specifically tailored settings (Argon gas flow: 1.6 L/min; precise mode, effect 1). The cells were treated in suspension on a 6-well cell culture plate in 700 µL DMEM at a distance of 7 mm. According to NIPP treatment, the controls were treated with argon gas alone (flow: 1.6 L/min) to exclude any specific alterations in cells and tissues due to argon gas. Human cervical tissue samples were treated by uniform motion and under constant wetting with Dulbecco’s phosphate buffered saline (DPBS) at a distance of 7 mm. Thermal damage was avoided. The control tissues were treated with argon gas alone and for 5 min at a distance of 7 mm (flow: 1.6 L/min) with constant motion and under constant wetting with DPBS to avoid drying effects. The control tissues thereby matched the epithelium initially located next to the NIPP-treated epithelium.

### 4.3. Live/Dead: PI/FDA Staining

To perform the live/dead assay with propidium iodide (PI, 72 µg/mL; Cat. no. P4170-10MG, Sigma-Aldrich, St. Louis, MO, USA) and fluorescein diacetate (FDA, 8 µg/mL; Cat. no. F1303, ThermoFischer Scientific, Waltham, MA, USA), cells were NIPP-treated for 30 s and cultured for 24 h in 6-well plates (150,000 cells per well). The staining of cells was performed for 15 min in the dark. FDA, a cell-permeant esterase substrate, is dependent on enzymatic activity of living cells to activate its fluorescence. The integrity of the cell membrane results in the intracellular retention and accumulation of the fluorescent FDA product. Due to the porous cell membrane of dead cells, the fluorescent dye PI can pass though, and intercalates the DNA. This enables the discrimination between living and dead cells. The cells were immediately analyzed by fluorescence microscopy after washing with PBS. For this, an inverted DMi8 light microscope (Leica, Wetzlar, Germany) with an integrated incubator was used. For every independent experiment (*n* = 3), the cells were measured in triplets.

### 4.4. Live/Dead: Guava ViaCount Assay

Guava ViaCount Reagent for Flow-Cytometry (Cat. no. 4000-0040, Merck, NJ, USA) was performed as recommended by the manufacturer. After CAP treatment for 30 s in 6-well plates (1.5 × 10^5^ cells per well) adherent cells were detached with 1% trypsin after 24 h of incubation. Supernatants were collected to avoid losing any detached dead cells. Washed once with PBS and resuspended 1:10 in Guava ViaCount Reagent, cells were immediately analyzed by flow cytometry of 3 × 10^3^ cells each with a Guava easyCyte Benchtop Flow Cytometer 8, 8HT (Merck). Numbers of vital and dead cells out of 3 independent experiments were determined by the Guava ViaCount Software (Merck).

### 4.5. Apoptosis: Caspase-Glo 3/7 Assay

The determination of apoptosis in 30 s NIPP-treated cells was performed using the luminescent Caspase-Glo 3/7 assay (Cat. no. G8090, Promega, Walldorf, Germany) to measure caspase-3 and -7 activities. The Caspase-Glo 3/7 assay was performed on adherent cells in 96-well plates (5000 cells per well) after 24, 48 and 72 h, as recommended by the manufacturer. Luminescence recording was performed by a Synergy 2 Multi-Mode Microplate Reader utilizing Microplate Data Collection and Analysis Software Gen5 (BioTek Instruments, Winooski, VT, USA). Each independent experiment was performed in triplicates. After the subtraction of the blank control, the luminescence intensities of the NIPP cells were normalized to the untreated cells (control). For every independent experiment, the cells were measured in triplets.

### 4.6. Proliferation Assay

Cellular proliferation was analyzed by cell counting using a CASY Cell Counter and Analyzer Model TT (Roche Applied Science). Per experiment, 5000 cells in suspension with 700 µL DMEM were NIPP-treated with different NIPP exposure times (5, 10, 30, 60, 90, and 120 s) on an uncoated cell culture plate. The cells were transferred onto 24-well cell culture plates (1 mL/well) and were cultured for 120 h. The adherent cells were detached by 0.25% trypsin-EDTA, followed by resuspension in a defined volume (200 µL) of CASYton solution (Roche Applied Science). The living cells were analyzed in duplicates for each sample within three independent experiments. The controls were treated with 4 L/min argon gas without plasma discharge for 120 s, respectively. For every independent experiment, the cells were measured in triplets.

### 4.7. Human Tissue Samples

After written informed consent of the patients was obtained, healthy tissue samples of non-keratinized squamous epithelium of the ectocervix uteri (Figure 3c) were taken under sterile conditions during vaginal hysterectomies indicated due to genital descent, at the Department of Women’s Health in Tübingen between April and October 2018. The scientific use of the tissue was approved by the Ethical Committee of the Medical Faculty of the Eberhardt-Karls-University Tübingen (649-2017BO2). The fresh tissue samples were transported in sterile DPBS (Dulbecco’s phosphate buffered saline) at 4 °C and were processed within one hour after removal. Tissue pieces of 3–5 mm × 10 mm × 20 mm were homogeneously NIPP- or control-treated for 2 and 5 min at a distance of 7 mm (flow: 1,6 l/min; precise mode, effect 1) with constant motion and under constant wetting with DPBS (Figure 3a). The tissues were incubated at 37 °C and 5% CO2 in a humidified atmosphere in keratinocyte growth medium 2 with SupplementMix and CaCl Solution (Cat. no. C-20011, C-39016, C-34005; PromoCell, Heidelberg, Germany). The tissues were cryopreserved with TissueTek (O.C.T.TM Compound; Sakura Finetek, Staufen im Breisgau, Germany) and freezing at −80 °C before being sectioned to 10 µm cross-sections. Prior to Raman measurement, the sections were redrawn with a PapPen (ImmEdge; Cat. no. H-4000; Vector Laboratories, Burlingame, CA, USA) and rinsed with DPBS five times.

### 4.8. Raman Imaging

A commercial Raman microscope (alpha 300 R; WiTec, Ulm, Germany) equipped with a green laser (532 nm) was used for Raman imaging. Burning was avoided by immersing the tissue samples in DPBS for the entire duration of the measurements to ensure the physiological conditions.

Raman measurements of the cryopreserved tissue samples were performed using a 63× dipping objective (NA 1.0; Carl Zeiss, Oberkochen, Germany), with a 0.5 µm scanning step size, and an integration time of 0.1 s per pixel within an area of 50 × 50 µm of both, the superficial and basal epithelial layer, using a.

### 4.9. Raman Image Analysis

All Raman images were pre-processed and further decomposed into spectral components using TCA, cosmic ray removal and baseline (shape) correction using Project FIVE software (WITec, Ulm, Germany).

The specific pixels containing significant amounts of lipids served to identify the spectral components in TCA. The pixels for each spectral component were averaged over the specific Raman image and demixed for glass and water background signal subtraction.

Mean-grey value intensities for each of the components in NIPP-treated and control tissues was semi-quantified out of 8-Bit using ImageJ 1.52a (Wayne Rasband, National Institute of Health) after adjusting charge-coupled device (CCD) count intensities and excluding the black areas (threshold, 5–255) for all images.

For PCA, only high-intensity pixels representing specific spectral lipid components were exported. Representative pixels were identified by sum intensities for lipid peaks (wavenumber: 2850 ± 5 rel. cm^−1^). A total of 50 spectra/pixels were randomly selected by MatLab R2018a (The MathWorks, Natick, MA 01760-2098, USA) for each patient and component, followed by cropping to 400–3000 rel. cm^−1^ and performing PCA.

### 4.10. Principal Component Analysis

The characterization of spectral differences within the different data sets by PCA (using Unscrambler × 10.5. (Camo, Oslo, Norway)) was previously described (Figure 3g) [39,40,41]. For each patient sample and treatment group a separate PCA was performed and the changes in score values were normalized to the respective control tissues to perform further statistical analysis between multiple donors. Hotelling’s T² test was used to exclude outliers. The vectors (PC1, explaining the main variance in spectral information and PC2, explaining the second variance in spectral information) represent the principal components (PC). Up to 7 PCs were calculated for every PCA using a nonlinear iterative partial least square (NIPALS) algorithm.

### 4.11. Statistical Analysis

Prism 6.0 (GraphPad, San Diego, CA, USA) was utilized for statistical analysis and comparisons. For image analysis, a Kruskal–Wallis test with Dunn’s multiple comparisons test was performed on each component and condition. For variance analysis (PCA), the Two-Way ANOVA and Sidak’s Multiple Comparison were performed. The data are expressed as mean ± standard deviation. P values of 0.05 or less were considered statistically significant.

## 5. Conclusions

The aim of this study was the evaluation of the effects of NIPP on cancer cell proliferation and viability, as well as the molecular mode of action. The results clearly indicate that NIPP treatment with non-thermally operated electrosurgical argon plasma devices is a promising treatment option for several diseases of human mucosa, in particular pre-cancerous and cancerous diseases. The significant cell effects are thereby comparable to conventional CAP sources.

## Figures and Tables

**Figure 1 cancers-12-00267-f001:**
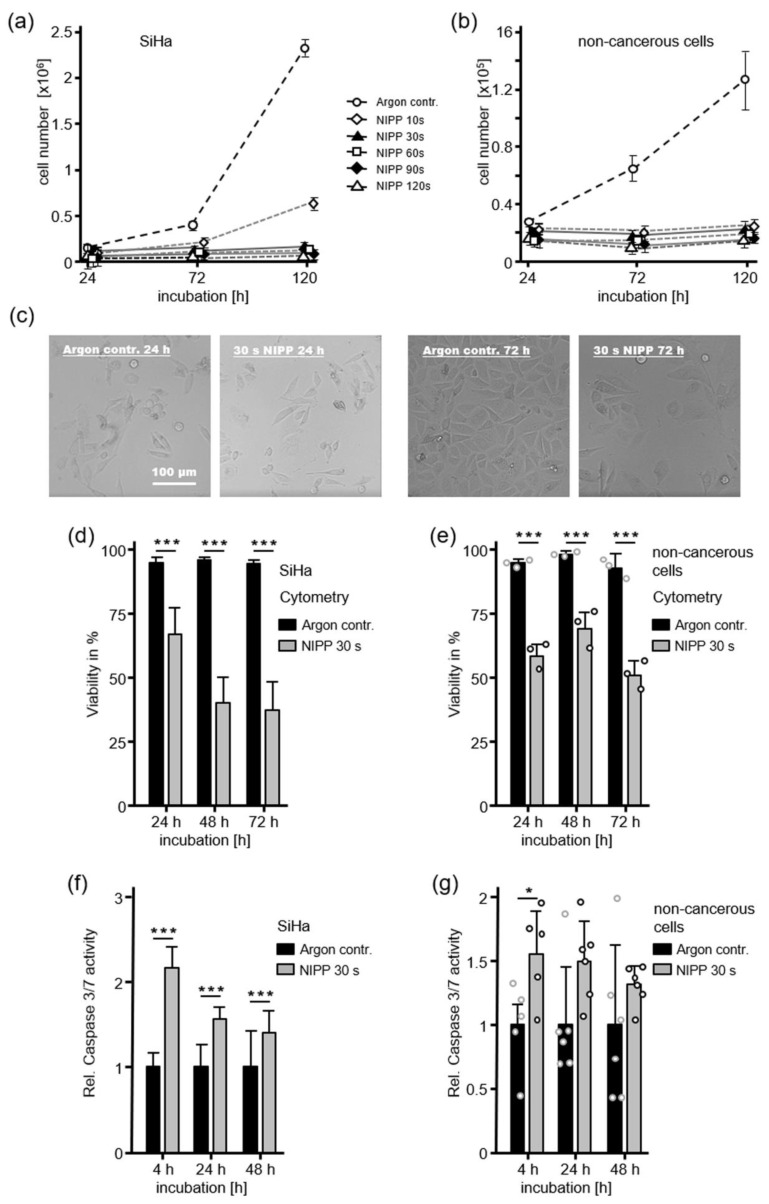
Non-invasive physical plasma (NIPP) has antiproliferative and cytotoxic effects on cervical cancer (CC) cells and primary cells from healthy cervical tissue. (**a**,**b**) After treatment with different doses of NIPP or argon gas, cell numbers dose-dependently decreased after 24, 72 and 120 h. (**c**) Surviving CC cells showed altered cell morphology after NIPP treatment. (**d**,**e**) In both cell types, NIPP-induced cytotoxicity was determined by staining with Guava ViaCount Reagent and subsequent flow cytometry. (**f**,**g**) Caspase-Glo 3/7 assay indicated apoptotic cell death induced by NIPP. This effect was higher on CC cells compared to cells from healthy cervical tissue. The results are expressed as the mean ± SD of at least three independent experiments. For non-cancerous cells, each independent experiment was performed with cells from a different donor. * *p* < 0.05 and *** *p* < 0.001, as determined by Student’s *t*-Test.

**Figure 2 cancers-12-00267-f002:**
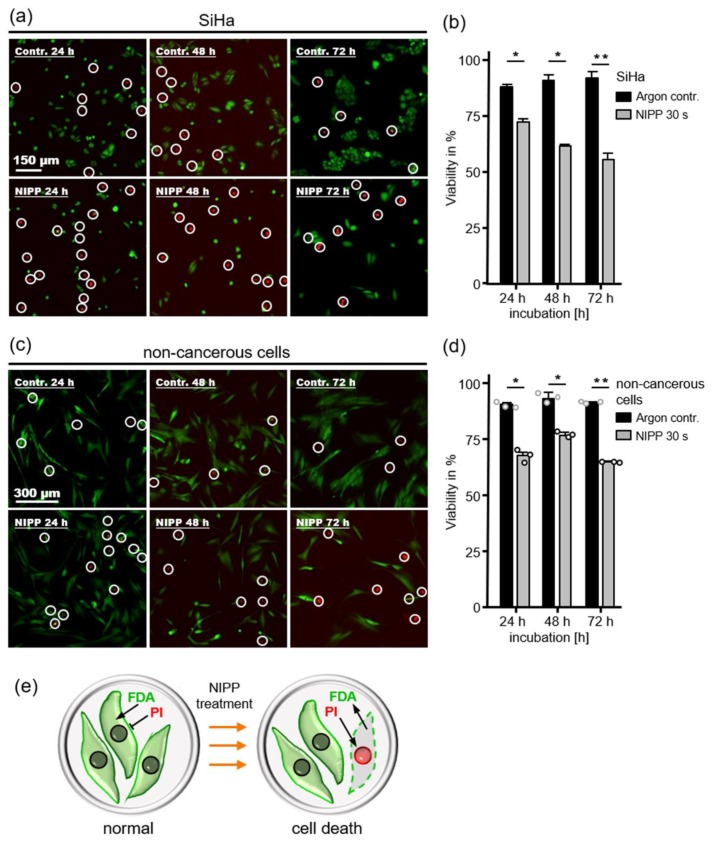
NIPP effects on cellular viability and membrane integrity of CC cells and primary non-cancerous cells from healthy cervical tissue. Cells were NIPP- or argon gas-treated for 30 s and analyzed after 24, 72 and 120 h. (**a**,**c**) Representative fluorescence microscopy after staining of native cells with PI and FDA. White circles indicate the presence of red stained cell nuclei. (**b**,**d**) Relative live/dead ratio by automatic counting of red and green fluorescent nuclei, using the image analysis software ImageJ compared to argon gas-treated controls. (**e**) Schematic functionality of FDA and PI staining in viable and dead cells. The results are expressed as the mean ± SD of three independent experiments. For non-cancerous cells, each independent experiment was performed with cells from a different donor. **p* < 0.05, ***p* < 0.01, as determined by Student’s *t*-Test.

**Figure 3 cancers-12-00267-f003:**
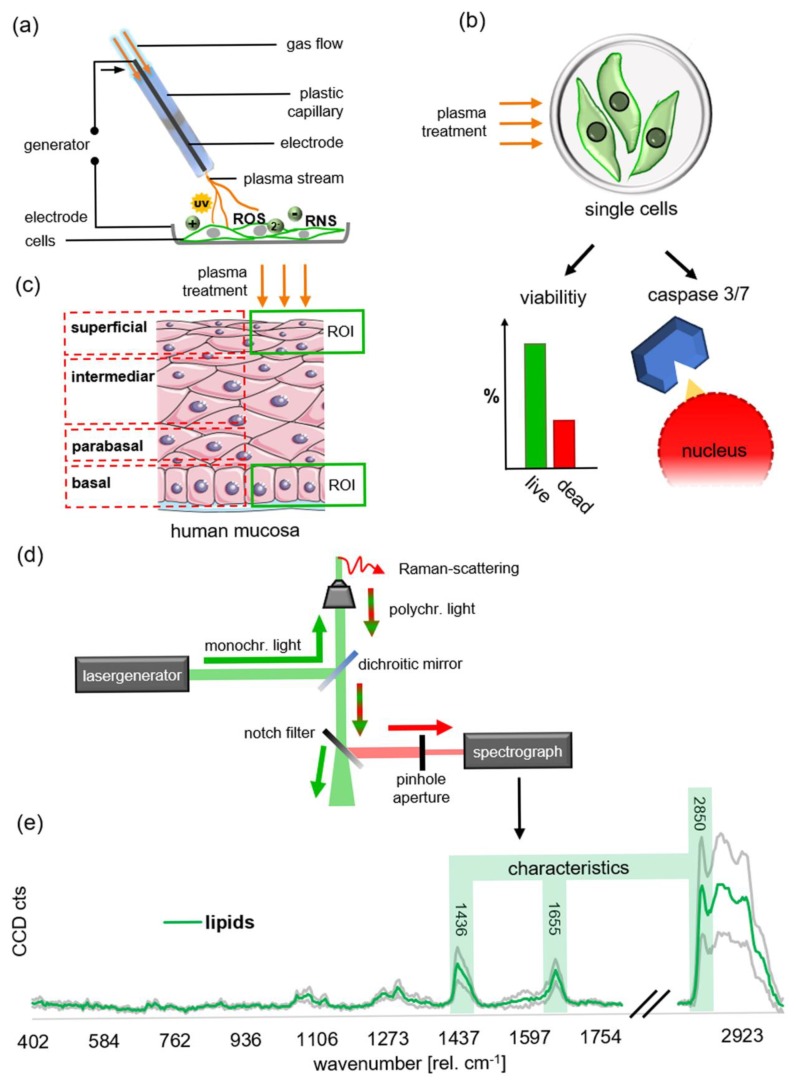
Schematic illustration of the experimental setup. (**a**) Setup of superficial tissue treatment with NIPP, utilizing the Vio3/APC3 (Erbe Elektromedizin); (**b**) CC single cell analysis via live/dead staining and Caspase3/7 assays; (**c**) Structure of stratified squamous epithelium in cervical tissue, modified based on https://creativecommons.org/licenses/by/3.0/deed.de. Green boxes designate the investigated regions of interest (ROI); (**d**) Schematic of the Raman microscope. (**e**) Representative Raman spectrum from untreated cervical control tissue. Wavelengths highlighted with green bars represent the characteristic composition of bands (at 1436, 1657, and the main band at 2850 rel. cm^−1^) utilized in this study to identify specifically lipid components out of the multiple Raman spectra obtained by Raman imaging, representing various biomolecules [27,28].

**Figure 4 cancers-12-00267-f004:**
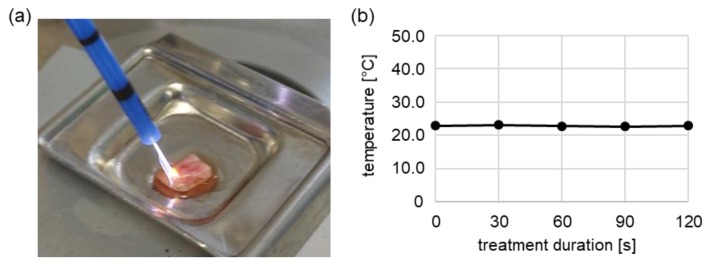
Superficial NIPP treatment of human cervix uteri. (**a**) Superficial treatment of cervical tissue with the Vio3/APC3. Laser thermographic assessment of tissue temperatures during dynamic superficial treatment of human tissue samples (**b**). The results are expressed as the mean ± SD.

**Figure 5 cancers-12-00267-f005:**
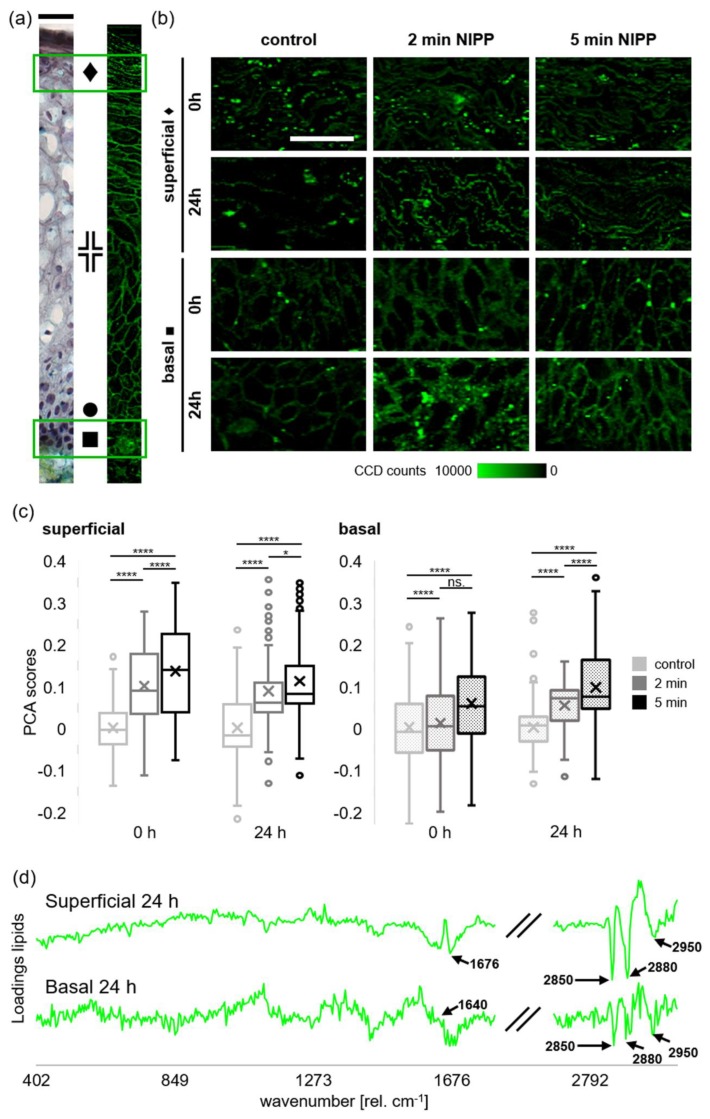
Raman imaging and molecular analysis of lipid components in NIPP-treated human cervical mucosa. (**a**) Staining by Movat Pentachrom histochemistry (left) and Raman imaging of lipid components (right) of native epithelial tissues; (**b**) ♦ = superficial, ╬ = intermediar, ● = parabasal, ■ = basal tissue layer. Raman images of lipid components (green) of NIPP treated and argon control samples analyzed by true component analysis (TCA). The scale bar equals 20 μm; (**c**) Boxplot analysis of principle component analysis (PCA) scores (*n* = 3) comparing lipid effects in superficial and basal cell layers respectively after 2 and 5 min of superficial NIPP- or argon control-treatment after 0 and 24 h of incubation; (**d**) Representative loading plots of superficial and basal lipid components based on respective PCA scores after 24 h of incubation. The results are expressed as the mean ± SD of PCA scores. * *p* < 0.05, **** *p* < 0.0001, as determined by Two-way ANOVA and Sidak’s multiple comparisons test.

**Table 1 cancers-12-00267-t001:** Identified characteristic Raman peaks [rel. cm^−1^] linked to lipid components and their molecular assignments according to literature.

Peaks (rel. cm^−1^)	Found in	Assignment	Reference
1169	NIPP	C=C stretch lipids	[29,30]
1306	NIPP	CH_3_/CH_2_ twisting or bending mode of lipid/collagen; Lipid/protein	[29,30]
1368	NIPP	ν_s_ (CH_3_) (phospholipids)	[29,31]
2850	control	ν_s_ CH_2_, lipids, fatty acids, CH_2_ symmetric	[29]
2880	control	CH_2_ asymmetric stretch of lipids	[29]
2910–2920	NIPP	C-H vibrations in lipids ν_as_ CH_2_, lipids, fatty acids; saturated and unsaturated fatty acids	[29,32]
2950	control	CH_3_ asymmetric stretch	[29]

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
