# Peer review of "Trans-Mucosal Efficacy of Non-Thermal Plasma Treatment on Cervical Cancer Tissue and Human Cervix Uteri by a Next Generation Electrosurgical Argon Plasma Device"

_cancers, 2020, doi:10.3390/cancers12020267_

Round 1

Reviewer 1 Report

The manuscript "Trans-mucosal Efficacy of Non-thermal Plasma Treatment ..." reports data from a study of plasma exposure effects of cervix tissue. These data are interesting for the development of innovative, promising (physical) plasma-based techniques in oncology. The manuscript should be published, possibly after minor revision to address the following comments, doubts and concerns.

Probably this paper will attract the interest of a broad readership, from the physics community (to which I belong) to the medicine community. For this reason, it is useful to describe more precisely what a "non-thermal" plasma is - I assume that it is a far-from-thermal equilibrium plasma, which contains a sufficient flow of energetic ions to drive the desired reactions while containing a modest overall amount of energy. The authors should also make clear the distinction between the energy per plasma particle and the overall energy contained in the plasma stream and to which the tissue is exposed, since sentences like "operated in a lower energy range" may leave some doubts to which energy they refer to.

The sentence "The thermal effect during tissue coagulation is based on the principal [sic] of energy conversion (electric energy is transformed into thermal energy)" is a rough, too generic description. The authors should clarify whether tissue heating is mainly due to direct energy deposition by the plasma stream or by induced current in the tissue (causing Joule heating), which may happen if the plasma stream is not electrically neutral.

About Raman spectra in Fig.5d): while the lines above 2800/cm seem to be prominent and show significant differences between superficial and basal samples, it is unclear whether any significant information can be obtained from the spectral region below 1700/cm. What is the origin of the global difference of the spectral profile (not limited to characteristic "lipid" lines) between the superficial and basal tissues? Is there a spectrum from the "control" (unexposed) tissue? A minor related point: what are the differences between the three spectral profiles in Fig.1e)? If they represent different exposure conditions, it seems that only the >2800/cm band is sensitive enough to provide an useful diagnostic.

In figure 4b), are statistical error bars contained in the dots used to represent data points?

Misprints: "principal" for "principle"; "vales" for "values"; "Contol tissues"; "elektrode" and "charcateristics" in the labels of Fig.1.

Author Response

please view at the attachment.

Reviewer 2 Report

The paper from Wenzel et al. deals with the effect of atmospheric non-thermal plasma treatment on cervical cancer cells. The topic is of extreme interest for the plasma medicine scientific community. The paper is well written. The proposed introduction is sufficiently exhaustive, which is a relevant point, in my opinion, when such a multidisciplinary field of research is considered. Figures are also clear, with only small typographic errors.

The adopted methods are rigorous and described with good detail. A variety of experimental conditions and analyses has been investigated along with sophisticated applications of experimental techniques, such as, for example, Raman imaging of solid cervical tissues, which gives important morphological and biochemical  information after plasma treatment.  An interesting dose dependent effect on the biochemical composition is found in both superficial and basal tissue layers.

Single cell analysis is also performed and, in my opinion, the Authors are right on their interpretation of the role of the indirect plasma activated medium on the observed effects.

On the basis of the given results, I agree with the Authors in proposing a next generation electrosurgical Argon plasma device as a promising method for the treatment of various diseases of human mucosa.

For all these reasons, I think that paper deserves publication in its present form.

Author Response

For all these reasons, I think that paper deserves publication in its present form.

Reviewer 3 Report

Publish as is.

Author Response

Publish as is.